# Amazing Combinatorial Creation: Acceptable Swap-Sampling for Text-to-Image Generation

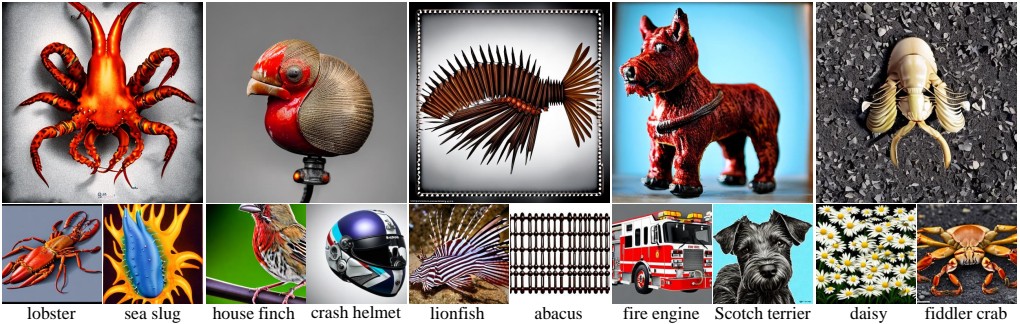

Figure 1: We propose a simple yet effective sampling method without any training to generate new and meaningful combinations from two given object texts in text-to-image synthesis. The original images of each object text using Stable-Diffusion2 (Rombach et al., 2022) are displayed in the bottom row, while the top row showcases the amazing combinations produced by our sampling algorithm.

## ABSTRACT

Exploring a machine learning system to generate meaningful combinatorial object images from multiple textual descriptions, emulating human creativity, is a significant challenge as humans are able to construct amazing combinatorial objects, but machines strive to emulate data distribution. In this paper, we develop a straightforward yet highly effective technique called *acceptable swap-sampling* to generate a combinatorial object image that exhibits novelty and surprise, utilizing text concepts of different objects. Initially, we propose a swapping mechanism that constructs a novel embedding by exchanging column vectors of two text embeddings for generating a new combinatorial image through a cutting-edge diffusion model. Furthermore, we design an acceptable region by managing suitable CLIP distances between the new image and the original concept generations, increasing the likelihood of accepting the new image with a high-quality combination. This region allows us to efficiently sample a small subset from a new image pool generated by using randomly exchanging column vectors. Lastly, we employ a segmentation method to compare CLIP distances among the segmented components, ultimately selecting the most promising image from the sampled subset. Our experiments focus on text pairs of objects from ImageNet, and our results demonstrate that our approach outperforms recent methods such as Stable-Diffusion2, DALLE2, ERNIE-ViLG2 and Bing in generating novel and surprising object images, even when the associated concepts appear to be implausible, such as *lionfish-abacus* and *kangaroo-pears* (see Figs. 1 and 5). Furthermore, during the sampling process, our approach without training and human preference is also comparable to PickScore (Kirstain et al., 2023) and HPSv2 (Wu et al., 2023) trained using human preference datasets. Anonymous Project page: `https://asst2i.github.io/anon/`

## 1 INTRODUCTION

Human creativity plays a crucial role in the innovative visual generation from textual concepts, known as text-to-image (T2I) synthesis. However, this task poses a significant challenge for most existing methods in computer vision and machine learning, including DALLE2 (Ramesh et al., 2022),

Stable-Diffusion2 (Rombach et al., 2022), and ERNIE-ViLG2 (Feng et al., 2023b). These methods aim to generate images that emulate a given training distribution (Elgammal et al., 2017), but they often lack the potential for novelty and surprise (Das & Varshney, 2022). Consequently, there is a need to develop machine learning systems with enhanced novel and surprising capabilities.

Recent efforts have primarily focused on compositional objects, aiming to directly generate new and intricate images by composing textual descriptions of multiple known objects. One example is the application of composable diffusion models (CDMs) (Liu et al., 2022b), which generate images containing multiple objects at specified positions. Another approach, known as Structure-Diffusion (Feng et al., 2023a), incorporates linguistic structures to generate image layouts that are plausible and do not omit any objects. Additionally, Custom-Diffusion (Kumari et al., 2023) enables the generation of new and reasonable compositions of multi-objects in previously unseen contexts. However, these compositional methods only produce generations with independent objects, lacking the element of novelty and surprise (see Fig. 1 (Feng et al., 2023a), Fig. 2 (Saharia et al., 2022), and Fig. 1 (Kumari et al., 2023)). This raises an important question: How can we produce a new and meaningful object by combining two object concepts, such as *"lionfish"* and *"abacus"*?

In this work, we present a novel technique called *acceptable swap-sampling* (ASS) that generates unique and surprising combinatorial objects by combining the prompt embeddings of two seemingly unrelated object concepts. Our approach comprises a text encoder, a swapping mechanism, an image generator, and an acceptable region using a CLIP metric (Radford et al., 2021). To start, the text encoder and image generator can be pretrained using state-of-the-art T2I models, such as Stable-Diffusion2 (Rombach et al., 2022). Initially, we obtain two original embeddings by inputting the prompts of two object texts into the text encoder. Then, two original images are generated by using these embeddings in the image generator. Next, we propose a swapping mechanism that interchange column vectors of the prompt embeddings. This operation results in a novel embedding, allowing the image generator to create a fresh and distinctive combinatorial image. Furthermore, we establish an acceptable region within which the newly created image likely exhibits a high-quality fusion of the two original concepts. This is achieved by carefully managing the CLIP distances between the newly creations and the two originally generations. The acceptable region allows us to efficiently sample a small subset of the newly created images from a pool of randomly exchanging column vectors. Lastly, we employ a segmentation method to compare CLIP distances among the segmented components, ultimately selecting the most promising combinatorial image from the sampled subset.

To demonstrate the feasibility of our ASS technique, we randomly select object pairs from ImageNet (Russakovsky et al., 2015). Experimental results demonstrate that our strategy can generate novel and surprising object images surpassing most SOTA T2I methods, including Stable-Diffusion2, DALLE2, ERNIE-ViLG2, and Bing, even when the associated object concepts appear implausible. For instance, in Fig. 1, the second column showcases an amazing and surprising artwork that abstractly combines characteristics of lionfish and an abacus. Overall, our contributions are summarized as follows:

- We propose a swapping technique that exchanges column vectors of prompt embeddings between two given object texts. This constructs a new prompt embedding that significantly deviates from the data distribution, resulting in novel combinatorial object generations.

- We define an acceptable region by controlling the CLIP distances between the newly generated image and the two originally created images. Within this defined range, the generated image using two original concepts likely exhibits their high-quality fusion.

- By combining the swapping technique with the acceptable region, we explore an acceptable swap-sampling approach to produce amazing combinatorial object image generations without any training. To the best of our knowledge, we are the pioneers in developing a machine learning system with exceptional combination capabilities in text-to-image synthesis.

- Experimental results demonstrate the effectiveness of our approach in generating previously unseen and unexpected object images in Figs. 1 and 5. Our method surpasses the object images generated by the current SOTA T2I techniques. More results, see the appendix F.

## 2 METHODOLOGY

In this section, we introduce an acceptable swap-sampling (ASS) approach for generating a new and surprising object image using two given concept texts, as shown in Fig. 2. It includes a swapping

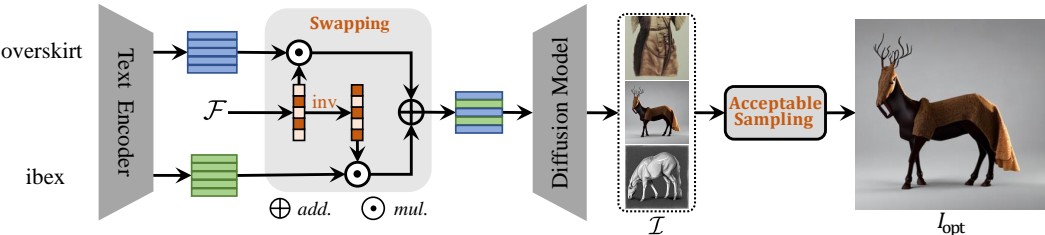

Figure 2: The pipeline of our acceptable swap-sampling method. Starting from text embeddings by inputting two given texts into the text encoder, we introduce a swapping operation to collect a set $\mathcal{F}$ of randomly swapping vectors for novel embeddings, then generate a new image set $\mathcal{I}$, and propose an acceptable region to build a sampling method for selecting an optimal combinatorial object image.

mechanism, an acceptable region and our ASS method. Before delving into the details of our method, we provide an overview of the unified generation process using the T2I models (*e.g.*, Stable-Diffusion2 (Rombach et al., 2022), DALLE2 (Ramesh et al., 2022), and Imagic (Saharia et al., 2022)).

**T2I:** For a text $t$ and its associated prompt $p$, a generated image is described as $G = \mathcal{G}(E)$, where $E = \mathcal{E}(p) \in \mathbb{R}^{h \times w}$ is a text encoder with dimensions $h$ and $w$, and $\mathcal{G}(\cdot) \in \mathbb{R}^{H \times W}$ is an image generator with dimensions $H$ and $W$.

Given a text pair $(t_1, t_2)$, we use the T2I model to generate their original images $I_1 = \mathcal{G}(\mathcal{E}(p_1))$ and $I_2 = \mathcal{G}(\mathcal{E}(p_2))$, where $p_1$ and $p_2$ are the prompts of $t_1$ and $t_2$, respectively. For example, for a text pair (*lobster*, *sea slug*), we use its prompt pair *(A photo of lobster, A photo of sea slug)*, to produce two images, please refer to the two below figures in the first column of the Fig. 1. Note that the text encoder $\mathcal{E}(\cdot)$ and the image generator $\mathcal{G}(\cdot)$ can be pretrained using Stable-Diffusion2 (Rombach et al., 2022), as our baseline. Alternatively, any other diffusion model can also be utilized in our approach.

## 2.1 DEVELOPING A SWAPPING MECHANISM TO GENERATE NEW COMPOSITE IMAGES

Following the generation process of the T2I model, we propose a swapping operation to mix well the prompt embeddings of a given prompt pair for a new image generation, as shown in the left part of Fig. 2. The swapping process is formalized as the following three steps:

- **Encoding** a prompt pair $(p_1, p_2)$ by using a text encoder,
$$E_1 = \mathcal{E}(p_1) \in \mathbb{R}^{h \times w} \quad \text{and} \quad E_2 = \mathcal{E}(p_2) \in \mathbb{R}^{h \times w}, \tag{1}$$

- **Swapping** their column vectors by using an exchanging vector $f \in \{0, 1\}^{w \times 1}$,
$$E_f = E_1 \text{diag}(f) + E_2 \text{diag}(1 - f) \in \mathbb{R}^{h \times w}, \tag{2}$$

- **Generating** a novel image by using an image generator,
$$I_f = \mathcal{G}(E_f) \in \mathbb{R}^{H \times W}, \tag{3}$$

where $\text{diag}(\cdot)$ is an operation to diagonalize a vector, and $f \in \{0, 1\}^{w \times 1}$ is a binary vector to swap the column vectors of the prompt embeddings $E_1$ and $E_2$. (The neural swapping in Appendix A)

**Why swapping column vectors of prompt embeddings** $E_1$ **and** $E_2$**?** The swapping process revolves around effectively and thoughtfully combining meaningful characteristics of the prompt embeddings $E_1$ and $E_2$. Firstly, a simple combination approach involves linear interpolation, represented by the purple line in Fig. 3, where a linear embedding space is spanned using the formula $\alpha E_1 + (1 - \alpha)E_2$, with $\alpha$ ranging between 0 and 1. However, this method often produces expected and unsurprising embeddings. For example, while Magicmix (Liew et al., 2022) employs linear interpolation to blend two concepts, such as creating a *corgi-alike coffee machine*, the results may appear somewhat unnatural and lack artistic merit. Secondly, to achieve a more surprising combination, an effective method is to swap the corresponding elements of $E_1$ and $E_2$. However, finding the swapping matrix can be computationally expensive due to its size. Moreover, extensive experimentation shows that when swapping their row vectors multiple times, the generated images often lack meaning.

Thirdly, certain experiments have demonstrated meaningful results by swapping the column vectors multiple times. Importantly, the resulting embedding $E_f$ created through column vector swapping is novel because it samples from a combination distribution that mixes the column embedding

distributions of $E_1$ and $E_2$. As shown in Figs. 1 and 5 (Stable-Diffusion2 and Ours), the difference between $E_f$ and $E_1 \& E_2$ is significant. Additionally, the combinational embedding differs from the embedding distribution of the entire dataset, as illustrated in Fig. 5 (Dataset-Retrieval and Ours).

## 2.2 ESTABLISHING AN ACCEPTABLE REGION FOR SAMPLING POTENTIALLY HIGH-QUALITY COMPOSITE IMAGES

Using the above swapping technique in Eqs. 1-3, a new image $I_f$ is created by combining the text prompts $p_1$ and $p_2$. In this context, we establish an acceptable region to sample potentially high-quality composite images depending on only the images, $I_1$ and $I_2$, generated using $p_1$ and $p_2$ as anchor points, excluding human priors. This region adheres to a fundamental sense:

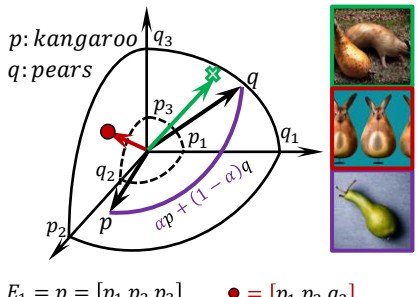

$E_1 = p = [p_1 \, p_2 \, p_3]$    ● $= [p_1 \, p_2 \, q_3]$
$E_2 = q = [q_1 \, q_2 \, q_3]$    ✖ $= [q_1 \, p_2 \, q_3]$

Figure 3: An example of swapping vectors based on *kangaroo* and *pears*. The red point and the green cross showcase meaningful and meaningless objects, respectively. The purple line shows unsurprising objects using liner interpolation.

*A new image $I_f$ can be considered of potential high-quality in a combinatorial sense*
*if it maintains an appropriate balance in distance from the anchor images $I_1$ and $I_2$.*

This distance must adhere to three key rules. The first rule entails maintaining a balance between the distances from $I_f$ to $I_1$ and from $I_f$ to $I_2$, demonstrating an equilibrium between them. The second rule highlights that a substantial separation between these distances signals a higher likelihood of generating content that is disordered, devoid of meaningful. The third rule underscores that a minimal distance indicates that the generated image closely resembles the input data, potentially lacking novelty and surprise. In mathematical terms, we can formalize them into two distance criteria:

- **Balancing Distances:** Achieving a balance between the distances $d(I_f, I_1)$ and $d(I_f, I_2)$ through an inequality involving a constant $\alpha$,

$$|d(I_f, I_1) - d(I_f, I_2)| \leq \alpha, \tag{4}$$

- **Controlling Bounds:** Constraining the upper bound of the average distance between $d(I_f, I_1)$ and $d(I_f, I_2)$ by using an inequality with a constant $\beta$,

$$d(I_f, I_1) + d(I_f, I_2) \leq 2\beta, \tag{5}$$

where $\alpha \geq 0, \beta \geq d(I_1, I_2)/2 \geq 0$, $d(I_1, I_2)$ is the distance between the anchor images $I_1$ and $I_2$, $|\cdot|$ is the absolute value function, and $d(a, b) \geq 0$ is a positive function to compute the distance between $a$ and $b$. These two criteria in the Eqs. 4 and 5 govern a region in within which the combinatorial image $I_f$ has the potential to be potential high-quality. Now, let's delve into a geometrical analysis.

**Geometrical Explanations.** Using the above criteria, we define the potential combinatorial space for the image $I_f$ using geometric shapes, specifically a hyperbola and an ellipse in Fig. 4. Within the context of Eq. 4, the combinatorial images are constrained by the hyperbola, expressed as $|d(I_f, I_1) - d(I_f, I_2)| = \alpha$, where $\alpha$ is a constant, and $I_1$ and $I_2$ are fixed points. This condition delineates the grey-shaded area. When $\alpha = 0$, it results in a balanced line perpendicular to the line connecting $I_1$ and $I_2$. Similarly, the inequality in Eq. 5 defines an upper boundary within an elliptical curve, given by $(d(I_f, I_1) + d(I_f, I_2)) = 2\beta$ with $I_1$ and $I_2$ as fixed points and a constant $\beta$. This condition defines the blue-shaded areas. When $\beta = d(I_1, I_2)/2$, it results in a single line connecting $I_1$ and $I_2$. When considering both the hyperbola and the ellipse simultaneously, the resulting image $I_f$ falls within the overlapping orange

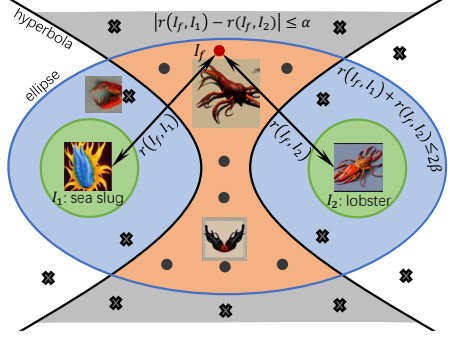

Figure 4: Geometrical visualization of the high-quality composite image's potential orange region by balancing the distances between $I_f$ and the anchor images $I_1, I_2$.

region, indicating its potential for creating high-quality blended images. The circle and fork points represent the acceptable and rejection images, respectively. It is important to note that while this acceptable region is not flawless, it provides a means to control over the qualities of the combinatorial image $I_f$ to a certain extent in Fig. 4.

Furthermore, the green areas indicate the distributions of the original images $I_1$ and $I_2$. They stand in contrast to the orange region shown in Fig. 4. This schematic diagram illustrates that the generated images within our acceptable region fall outside the data distribution (out-of-distribution), as demonstrated by comparing our created images with their the dataset retrievals presented in Fig. 5.

## 2.3 ACCEPTABLE SWAP-SAMPLING

Here, by combining the swapping technique with the acceptable region, we propose an acceptable swap-sampling method to sample a promising blend image $I_f$ on the prompt pair $(p_1, p_2)$. We first generate a set of $N$ random swapping vectors $\mathcal{F}$, and correspondingly produce an image set $\mathcal{I}$ using Eqs. 1-3. Note that $f \in \mathcal{F}$ corresponds to $I_f \in \mathcal{I}$ one by one. Depending on the prompts $p_1, p_2$, and their generated images $I_1, I_2$, we introduce a coarse-to-fine sampling method as follows.

---
**Algorithm 1** Acceptable Swap-Sampling (ASS).

---
1: **input:** prompt pair $(p_1, p_2)$, their images $I_1, I_2$;
2: **initialize:** $\theta = 0.05$, $\overline{\alpha} = 0.4$, $\overline{\beta} = 0.1$, $N = 200$;
3: Generate a set $\mathcal{F}$ of $N$ randomly swapping vectors;
4: Produce an image set $\mathcal{I}$ using Eqs. 1-3 and $f \in \mathcal{F}$;
5: Sample a coarse subset $\mathcal{I}_{\text{coarse}}$ using Eq. 6 with $\theta$;
6: Sample a fine subset $\mathcal{I}_{\text{fine}}$ using Eq. 7 with $\overline{\alpha}, \overline{\beta}$;
7: Choose the optimal image $I_{\text{opt}}^{(p_1,p_2)}$ using Eq. 8;
8: **output:** $I_{\text{opt}}^{(p_1,p_2)}$.

---

**Coarse Sampling Using Semantic Distance.** We select a coarse subset $\mathcal{I}_{\text{coarse}}$ from the set $\mathcal{I}$ by using a semantic balance, similar to the Eq. 4. This helps us maintain a balanced semantic content between the image $I_f$ and the text prompts $p_1$ and $p_2$. The coarse sampling is defined as follows:

$$\mathcal{I}_{\text{coarse}} = \{I_f \mid |d(I_f, p_1) - d(I_f, p_2)| \leq \theta, \ I_f \in \mathcal{I}\}, \tag{6}$$

where $\theta$ is a width threshold of the semantic balance area. It is set to $\theta = 0.05$ in this paper.

**Fine Sampling Using Image Distances.** We further choose a fine subset $\mathcal{I}_{\text{fine}}$ from $\mathcal{I}_{\text{coarse}}$ by using the acceptable region in Eqs. 4 and 5, ensuring a balanced relationship among $I_f, I_1$ and $I_2$. This leads to potential high-quality combinations. The fine sampling is expressed as:

$$\mathcal{I}_{\text{fine}} = \{I_f \mid |d(I_f, I_1) - d(I_f, I_2)| \leq \alpha \ \& \ d(I_f, I_1) + d(I_f, I_2) \leq 2\beta, \ I_f \in \mathcal{I}_{\text{coarse}}\}. \tag{7}$$

where $\alpha = d_{\lceil |\mathcal{I}_{\text{coarse}}| \cdot \overline{\alpha} \rceil}^d$, $\beta = d_{\lceil |\mathcal{I}_{\text{coarse}}| \cdot \overline{\beta} \rceil}^s$, $0 \leq \overline{\alpha} \leq 1$, $0 \leq \overline{\beta} \leq 1$, $\lceil \cdot \rceil$ denotes rounding up to the nearest integer, $|\mathcal{I}_{\text{coarse}}|$ represents the cardinality of the set $\mathcal{I}_{\text{coarse}}$, and $d_i^d$ and $d_i^s$ refer to the $i$-th element of the descendingly sorted sets $\mathcal{D}^d = \{d^d = |d(I_f, I_1) - d(I_f, I_2)|, I_f \in \mathcal{I}_{\text{coarse}}\}$ and $\mathcal{D}^s = \{d^s = d(I_f, I_1) + d(I_f, I_2), \ I_f \in \mathcal{I}_{\text{coarse}}\}$, respectively. In this paper, we set $\overline{\alpha} = 0.4$, and $\overline{\beta} = 0.1$. The subset $\mathcal{I}_{\text{fine}}$ indicates images that may exhibit excellent mixing characteristics, denoted as black points within the orange region in Fig. 4.

**Choosing the Optimal Image Using Segmentation Methods.** Unfortunately, $\mathcal{I}_{\text{fine}}$ is not directly employed for the ideal selection. Instead, we adopt the Segment Anything Model (SAM) (Kirillov et al., 2023) to enhance the visual semantic components, thereby facilitating the selection of the optimal combinatorial image. The final selection is made by maximizing the following objective:

$$I_{\text{opt}}^{(p_1,p_2)} = \arg\max_{I_f \in \mathcal{I}_{\text{fine}}} \{r(I_f, I_1, I_2)\} \ \text{with} \ r(I_f, I_1, I_2) = (s(I_f, I_1) + s(I_f, I_2))/2, \tag{8}$$

where $s(I_f, I_i) = \frac{1}{|\mathcal{C}_i| \times |\mathcal{C}_f|} \sum_{c_f \in \mathcal{C}_f, c_i \in \mathcal{C}_i} d(c_f, c_i)$, $\mathcal{C}_i = \text{SAM}(I_i)(i = 1, 2)$, $\mathcal{C}_f = \text{SAM}(I_f)$, and SAM$(I)$ represents a collection of segmented components extracted from the image $I$ using SAM (Kirillov et al., 2023). Using Eq. 8, we identify the optimal image from the $\mathcal{I}_{\text{fine}}$, as depicted by the red point in Fig. 4. This process culminates in the creation of a combinatorial object image that is both promising and amazing. Overall, our acceptable swap-sampling process is outlined in **Algorithm 1**.

Note that in this paper, we define the distance function as $d(a, b) = \cos(\phi(a), \phi(b))$, where $\cos(\cdot, \cdot)$ represents a cosine similarity function, and $\phi(\cdot)$ corresponds to the CLIP model (Radford et al., 2021).

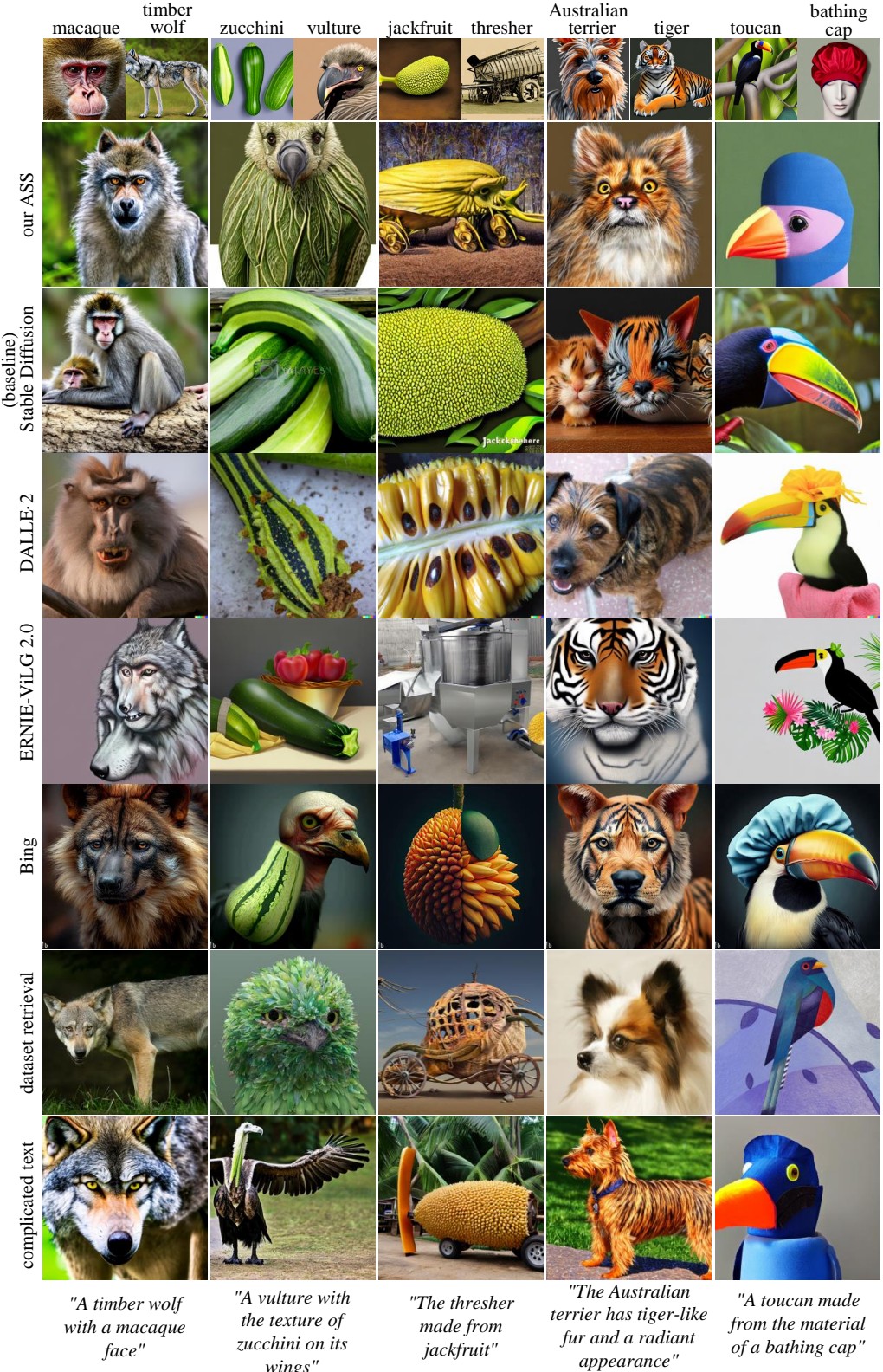

Figure 5: Visual comparisons of combinatorial object generations. We compare our ASS method with the SOTA T2I models, Stable-Diffusion2 (Rombach et al., 2022), DALLE2 (Ramesh et al., 2022), ERNIE-ViLG2 (Baidu) (Feng et al., 2023b) and Bing (Microsoft). Our findings indicate that our model demonstrates greater creative potential compared to these counterparts. Furthermore, our results exhibit significant dissimilarity from images retrieved from the LAION-5B dataset (Schuhmann et al., 2022) and the complex text generations in the last two rows. This directly illustrates that our approach has the ability to generate out-of-distribution images.

## 3 EXPERIMENTS

### 3.1 EXPERIMENTAL SETTINGS

**Dataset.** To showcase the power of combinational creativity in T2I synthesis, we have curated a novel dataset comprising prompt pairs. We leveraged the vast vocabulary of ImageNet (Russakovsky et al., 2015), consisting of 1,000 categories, to form our text set. From this collection, we randomly selected two distinct words to construct each prompt pair, such as *macaque-timber wolf*. Our dataset encompasses a total of 5075 prompt pairs, representing a fraction of the possible combinations.

**Sampling Settings.** For the sampling process, the text encoder $\mathcal{E}(\cdot)$ and the image generator $\mathcal{G}(\cdot)$ were pretrained using the CLIP model with ViT-L/14@336p backbone (Radford et al., 2021) and Stable-Diffusion2 (Rombach et al., 2022), respectively. This same CLIP model (Radford et al., 2021) was also employed in the distance function. The Segment Anything Model (SAM) (Kirillov et al., 2023) served as the pre-trained segmentation method. We conducted our experiments using four NVIDIA GeForce RTX 3090 GPUs, with a batch size of 64 per GPU.

**Evaluation metrics.** We assess our method on (1) text-alignment by calculating the cosine similarity between the image and the "hybrid of [prompt1] and [prompt2]" and (2) image-alignment by computing the average cosine similarity among the images generated by each prompt. By considering (1) and (2) together, we can determine if the image contains content from both prompts and assess the fusion effect. We also conduct a user study to evaluate the combined creativity of our approach.

### 3.2 MAIN RESULTS

We conducted a comprehensive comparison of our ASS method with four prominent Text-to-Image (T2I) models (*i.e.*, Stable-Diffusion2 (Rombach et al., 2022), DALLE2 (Ramesh et al., 2022), ERNIE-ViLG2 (Feng et al., 2023b) and Bing). Additionally, we compare with the sampling results obtained through PickScore (Kirstain et al., 2023) and HPS-v2 (Wu et al., 2023), which fine-tuned their CLIP models using human preference datasets. Furthermore, we also present a user study demonstrating the remarkable creative potential of our method when used in combination.

**Comparison with the SOTA T2I models.** Figs. 1 and 5 showcase several examples of image generation achieved through those models using prompt pairs. We make the following four observations. First, our model has the capability to create novel and surprising species that have never been seen before in real life. For instance, *kangaroo-pears* depicts the creative combination of kangaroos with the shape of pears, and *sunflower-orange* showcases oranges with sunflower-style segments in Fig. 1. Second, compared to the SOTA T2I models, our model exhibits a stronger ability to generate a creative object by inputting two different objects. Although the images generated by

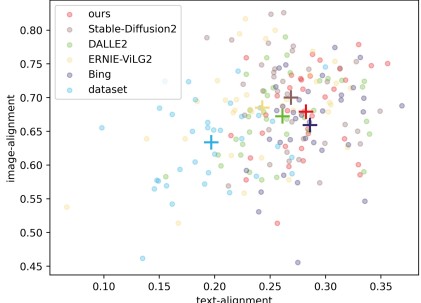

Figure 6: Text- and image-alignments.

other models are colorful and rich in detail, they do not fully display the mixed features of the two objects, as seen in *macaque-timber wolf* and *zucchini-vulture* in Fig. 5. Third, to evaluate the out-of-distribution generation ability of our model, we conducted a retrieval on the entire LAION-5B dataset (Schuhmann et al., 2022) to find the most similar image in the last second row of Fig. 5. By comparing our created images with the retrieved images, we found that they significantly differ from the retrieved ones, highlighting the distinctiveness of our model's output. Fourth, when comparing the images generated using intricate text descriptions, Stable-Diffusion2 still struggles to create plausible compositions, such as *A timber wolf with a macaque face* and *A vulture with the texture of zucchini on its wings* in the final row of Fig. 5. More visual comparisons, please refer to Appendixes B and F.

We conducted computations for *text-alignment* and *image-alignment* on thirty prompt pairs. The results obtained are plotted in Fig. 6. Our method (red plus) achieves a superior balance between text and image alignments in terms of mean performance. This indicates that the images generated by our method are more likely to blend both semantic and content information from the two prompts.

**Comparison with the evaluation sampling methods.** To assess the human-like superiority of our ASS method, we employed PickScore (Kirstain et al., 2023) and HPSv2 (Wu et al., 2023) to calculate

evaluation scores for selecting the optimal image with the highest score from the new image set $\mathcal{I}$. Subsequently, we utilized these metrics to evaluate the optimal images, and the findings are presented in Table 1. Our results indicate that our ASS method closely rivals the performance of PickScore and HPSv2. Notably, our method operates independently of human intervention, except for the selection of hyper-parameters using HPSv2. In contrast, both PickScore and HPSv2 rely on human performance datasets to fine-tune the CLIP models for T2I model evaluation. This demonstrates the robust capability of our ASS method to achieve performance comparable to

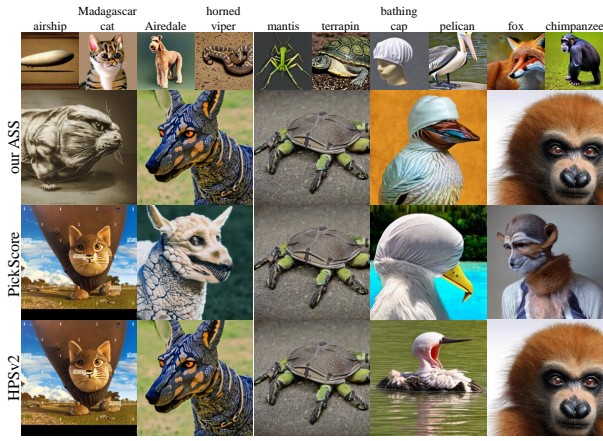

Figure 7: Sampling visualizations compared our ASS with the SOTA HPSv2 and PickScore.

that of humans. Furthermore, the sampling examples are illustrated in Fig. 7. Our ASS method excels in generating superior combinatorial images compared to both HPSv2 and PickScore.

Table 1: Quantitative comparisons.

| Models | PickScore | HPS-v2 | Our ASS |
|---|---|---|---|
| PickScore ↑ | 0.207 | 0.202 | 0.200 |
| HPSv2 ↑ | 0.246 | 0.253 | 0.242 |

Table 2: User study of combinational image creations.

| Models | Our ASS | SD2 (baseline) | DALLE2 | ERNIE-ViLG2 | Bing |
|---|---|---|---|---|---|
| Vote ↑ | **658** | 98 | 59 | 49 | 196 |

**User Study.** We conducted a user study to evaluate the combinational creativity of our model compared to four other T2I methods. Each user was presented with 10 pairs of prompts to vote on, resulting in a total of 1,060 votes from 106 users. The voting results are summarized in Table 2 and Appendix C. Our model received the highest number of votes, with 62% of users expressing that our model generates more creative samples. Additionally, 18.5% of users expressed interest in the Bing model, while the DALLE2 model received only 5% of the votes. Stable-Diffusion2, received 9% of the votes, while only 4.6% of users preferred ERNIE-ViLG2.

**Parameter Analysis.** We determined the parameters $\theta$ in Eq. 6, and $\overline{\alpha}$, and $\overline{\beta}$ in Eq. 7 using 20 text pairs. To begin, for each prompt pair $(p_1, p_2)$, we produce an image set $\mathcal{I}$ by randomly generating a set $\mathcal{F}$ consisting of $N = 200$ swapping vectors. From $\mathcal{I}$, we coarsely select a subset $\mathcal{I}_{coarse}$ using Eq. 6, and the parameter $\theta$ is set to 0.05 by choosing the best average HPSv2 score (Wu et al., 2023) in Table 3. This reudces the size of $\mathcal{I}_{coarse}$ approximates to 150. Next, we finely choose a subset $\mathcal{I}_{fine}$ from $\mathcal{I}_{coarse}$ using Eq. 7. We set that $\overline{\alpha} = 0.4$ and $\overline{\beta} = 0.1$ by selecting the best average HPSv2 score in Table 4. This reduces the size of $\mathcal{I}_{fine}$ to around 10. Finally,

Table 3: Parameter analysis with $\theta$ using average HPSv2 scores of 20 text pairs. $+\infty$ represents all sampling images.

| $\theta$ | 0.01 | 0.02 | **0.05** | 0.1 | $+\infty$ |
|---|---|---|---|---|---|
| HPSv2 | 0.2444 | 0.2451 | **0.2458** | 0.2392 | 0.2361 |

Table 4: Parameter analysis with $\overline{\alpha}$ and $\overline{\beta}$ using average HPSv2 of 20 text pairs.

| $\overline{\alpha}\backslash\overline{\beta}$ | 0 | 0.2 | **0.4** | 0.6 |
|---|---|---|---|---|
| 0 | 0.241 | 0.231 | 0.230 | 0.223 |
| **0.1** | 0.231 | 0.242 | 0.243 | 0.239 |
| 0.2 | 0.242 | 0.230 | 0.237 | 0.224 |
| 0.3 | 0.240 | 0.231 | 0.235 | 0.225 |

we obtain the optimal image $I_{opt}^{(p_1,p_2)}$ by maximizing the problem in Eq. 8. For illustrations of the sampled images using different $\theta$, $\overline{\alpha}$, and $\overline{\beta}$, please refer to Appendix D.

## 4 RELATED WORK

**Text-to-image (T2I) synthesis** has garnered increasing attention in recent years due to its remarkable progress (Zhu et al., 2019; Liao et al., 2022; Yu et al., 2022). Notably, diffusion models (Ho et al., 2020; Kawar et al., 2022; Liu et al., 2022a; Zhao et al., 2023; Song et al., 2023) combined with CLIP models (Radford et al., 2021) have shown great promise in T2I synthesis. For instance, CLIPDraw (Frans et al., 2022) employs only the CLIP embedding to generate sketch drawings. DALLE2 (Ramesh et al., 2022) introduces a diffusion decoder to generate images based on text concepts. Stable-Diffusion (Rombach et al., 2022), which performs on the latent diffusion space, has emerged as the most popular choice due to its open-source nature and ability to save inference time.

**Compositional T2I** primarily focuses on generating new and complex images by combining multiple known concepts. These concepts can be composed in various ways, including object-object, object-color and shape, object conjunction and negations, object relations, attributes, and modifying sentences by words. Some notable approaches in this field include object-object compositions (Liu et al., 2022b; Feng et al., 2023a; Kumari et al., 2023), image-concept compositions such as subject-context, segmentation-text, and sketch-sentence (Park et al., 2021; Du et al., 2020; Liu et al., 2022b; Chefer et al., 2023; Li et al., 2022; Cong et al., 2023; Gal et al., 2023; Hertz et al., 2023; Orgad et al., 2023; Ruiz et al., 2022; Brooks et al., 2023; Avrahami et al., 2023). However, a common limitation of these compositional T2I models is that they often generate images based solely on the compositional text descriptions, which restricts their creativity. Moreover, the existing object-object composition approaches (Liu et al., 2022b; Feng et al., 2023a; Kumari et al., 2023) tend to produce images with independent objects. Recent Magicmix (Liew et al., 2022) employs linear interpolation to merge distinct semantic images and text, aiming to produce novel conceptual images. But the outcomes may sometimes exhibit an unnatural blend and lack artistic value. In contrast, we propose an innovative sampling method that enables effective information exchange between two object concepts, leading to the creation of captivating composite images.

**Creativity** encompasses the ability to generate ideas or artifacts across various domains, including concepts, compositions, scientific theories, cookery recipes, and more (Boden, 2004; 1998; Maher, 2010; Cetinic & She, 2022; Hitsuwari et al., 2023). Recently, there has been significant research exploring the integration of creativity into GANs (Goodfellow et al., 2014; Ge et al., 2021) and VAEs (Kingma & Welling, 2014). For instance, CAN (Elgammal et al., 2017) extend the capabilities of GANs to produce artistic images by maximizing deviations from established styles while minimizing deviations from the art distribution. CreativeGAN, systematically modifies GAN models to synthesize novel engineering designs (Nobari et al., 2021b;a). Additionally, CreativeDecoder (Das et al., 2020; Cintas et al., 2022) enhances the decoder of VAEs by utilizing sampling, clustering, and selection strategies to capture neuronal activation patterns. Recent works (Boutin et al., 2022; 2023) design a method to assess one-shot generative models in approximating human-produced data by examining the trade-off between *recognizability* and *diversity* (measured by standard deviation). Unlike these methods, our approach introduces a swapping mechanism to to enhance the generation of novel combinational object images, along with a defined region for accepting high-quality combinations.

**Out-of-Distribution (OOD)** is closely related to our work as we generate creative object images that lie outside the data distribution. However, existing OOD techniques primarily concentrate on detection tasks (Shen et al., 2021; Ye et al., 2022) through disentangled representation learning (Träuble et al., 2021), causal representation learning (Shen et al., 2022; Khemakhem et al., 2020), domain generalization (Zhou et al., 2020a;b), and stable learning (Xu et al., 2022). In contrast, our focus is on OOD generation, where we aim to create meaningful object images. While (Ren et al., 2023) also incorporate an OOD generation step for the ODD detection task, they primarily address scenarios where the input distribution has shifted. In our approach, we generate a novel distribution by blending the embedding distributions of the input data.

## 5 CONCLUSION AND LIMITATION

**Conclusion.** We have incorporated a simple sampling method into text-to-image synthesis, proposing an acceptable swap-sampling schema to generate meaningful objects by combining seemingly unrelated object concepts. Our first idea involves a swapping process that exchanges important information from two given prompts, resulting in the creation of fresh object images that go beyond the original data distribution, thereby enhancing novelty. Additionally, we introduce an acceptable region based on the CLIP metric, which balances the distance among the given prompts, original image generations, and our creations to sample high-quality combinatorial object images from this pool of fresh object images. We futher employ the segment anything model to enhance the visual semantic components to select the optimal combinatorial image. Experimental results demonstrate that our approach surpasses popular T2I models in terms of generating creative combinatorial objects.

**Limitation.** Our method, despite its strengths, has a limitation: the acceptable region can sometimes result in non-meaningful or chaotic images. The high-quality acceptable area in an unsupervised manner continues to pose a challenging problem that necessitates further investigation. We have included the failure examples in the appendix E for reference.

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

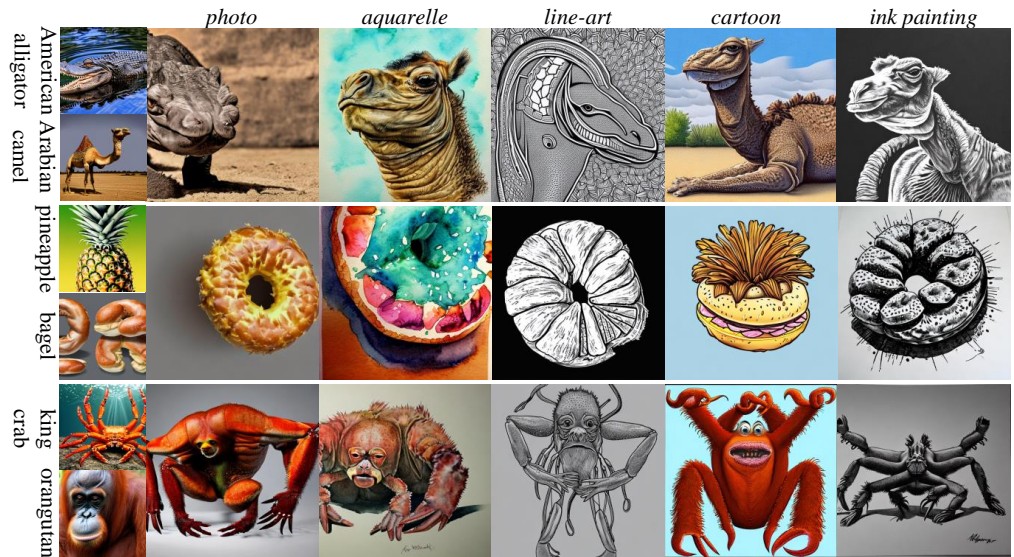

Figure 8: Generalizations using four different styles including *aquarelle*, *line-art*, *cartoon*, and *ink painting*. It can observe that our results are still novel and surprising.

## A    LEARNING A NEURAL SWAPPING NETWORK

In this subsection, we learn a neural swapping network to generate the meaningful combinatorial object images with different styles. The swapping part in the subsection 2.1 is rewritten as follows:

**Swapping** their column vectors by learning a neural swapping vector,

$$E_f = E_1 \text{diag}(f) + E_2 \text{diag}(1 - f) \quad \text{and} \quad f = S(\text{cat}(E_1, E_2); \psi) \in \{0, 1\}^{w \times 1}, \tag{9}$$

where $f = S(\text{cat}(E_1, E_2); \psi) \in \{0, 1\}^{w \times 1}$ is a neural swapping network from the concatenated embedding, $\text{cat}(E_1, E_2)$, to a binary output that consists of three convolutional layers and two fully connected layers with the parameter $\psi$. The architecture of the swapping network consists of three convolutional layers with a $3 \times 3$ kernel size, followed by two fully connected layers.

**Training Loss.** Using our ASS method in **Algorithm 1**, we can get the optimal combinatorial image $I_{\text{opt}}^{(p_1, p_2)}$, and then find its swapping vector $f_{\text{opt}}^{(p_1, p_2)}$. Based on the neural swapping network $f = S(\text{cat}(E_1, E_2); \psi)$ in Eq. 9 and the optimal swapping vector $f_{\text{opt}}^{(p_1, p_2)}$, the training loss is defined as:

$$L = \frac{1}{|\mathcal{P}|} \sum_{(p_1, p_2) \in \mathcal{P}} \|f_{\text{opt}}^{(p_1, p_2)} - S(\text{cat}(E_1, E_2); \psi)\|_2, \tag{10}$$

where $\mathcal{P}$ is a set of the prompt pairs $(p_1, p_2)$ corresponding to the text pairs $(t_1, t_2)$, and $|\mathcal{P}|$ represents the cardinality of the set $\mathcal{P}$. To ensure training stability, we employed the RMSprop (Hinton et al., 2012) optimizer.

**Generalizations using different styles.** To evaluate the generalizations of our model, we expand its capabilities to four additional styles: *aquarelle*, *line-art*, *cartoon*, and *ink painting* for text-to-image generation. We achieve this without the need for excessive sampling and training. The results in Fig. 8, demonstrate that our model maintains its impressive creativity by generating novel species across these different styles, such as *(orangutan, king crab)*.

## B    COMPARISON WITH MAGICMIX

In this section, we conduct a comparative analysis of our findings against those of Magicmix (Liew et al., 2022). It is worth noting that Magicmix is not currently available as open-source software. Consequently, we utilized an unofficial implementation, which can be found in (daspartho, 2022).

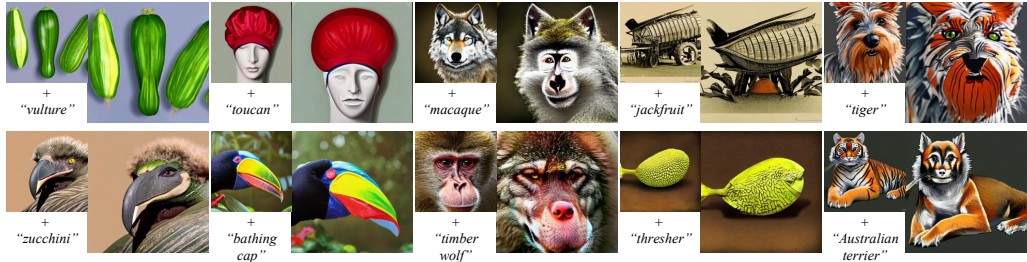

Figure 9: Generalizations using unofficial code (daspartho, 2022) of Magicmix with prompt-pairs in Figure 5.

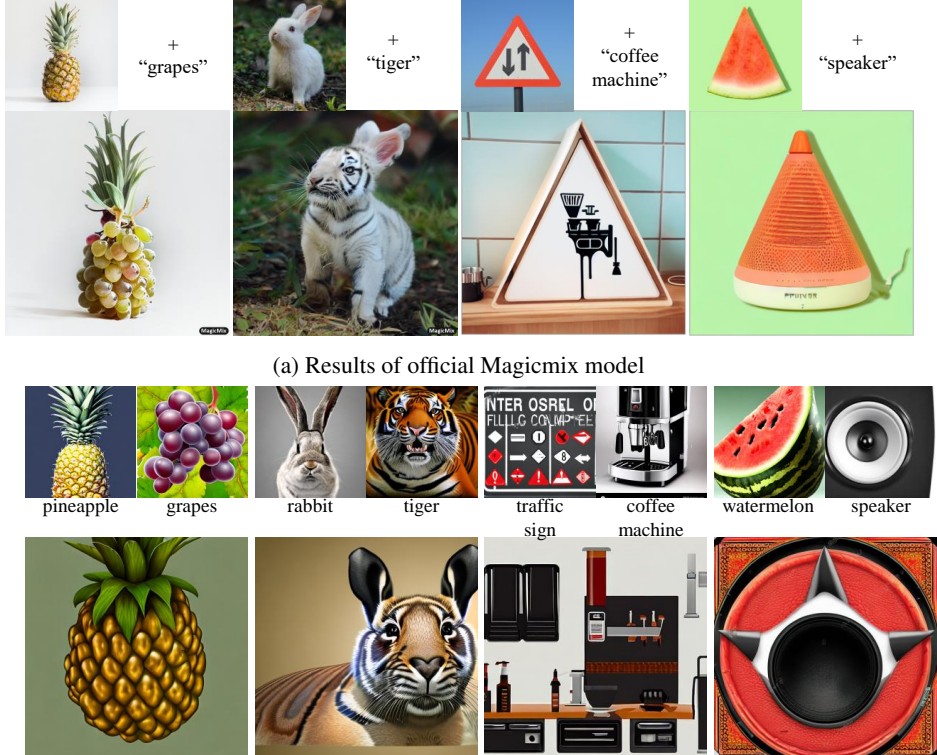

(a) Results of official Magicmix model

(b) Results using our ASS model with same prompt pairs of Fig. 10a.

Figure 10: Comparisons with Magicmix (Liew et al., 2022) with Our ASS using same prompts.

To initiate our investigation, we employed the Magicmix method to blend the five sets of word pairs depicted in Figure 5. Given that Magicmix operates with image-text inputs, we executed a total of 10 experiments for each pair of prompts. The outcomes of these experiments are presented in Figure 9.

In pursuit of official results for Magicmix, we acquired samples from its website (referenced as (Liew et al., 2022)), illustrated in Figure 10a. Subsequently, employing identical prompt-pairs, we employed our ASS model to generate images, as depicted in Figure 10b. These outcomes highlight a significant distinction between Magicmix and other component fusion generation models compared to our ASS model. Our primary objective is not mere imitation; instead, we aim to produce a greater number of Out-Of-Distribution objects based on provided prompts.

## C  USER STUDY

In this section, we delve into a more comprehensive explanation of our user study. In addition to the five result categories showcased in Figure 5, we have included an additional set of five prompt-pair groups, as illustrated in Figure 11, to facilitate our User Study. Within these ten queries, we

Table 5: The proportion of each option in each question in the User Study.

| options(Models) 
 prompt-pairs | A(our ASS) | B(baseline) | C(DALLE·2) | D(ERNIE-ViLG2) | E(Bing) |
|---|---|---|---|---|---|
| lionfish and abacus | 45.28% | 13.21% | 10.38% | 18.87% | 12.26% |
| lobster and sea slug | 62.26% | 12.26% | 12.26% | 3.77% | 9.43% |
| kangaroo and pears | 64.15% | 20.75% | 5.66% | 4.72% | 4.72% |
| sunflower and orange | 75.47% | 15.09% | 1.89% | 2.83% | 4.72% |
| macaque and timber wolf | 75.47% | 2.83% | 2.83% | 3.77% | 15.09% |
| Australian terrier and tiger | 51.89% | 12.26% | 1.89% | 3.77% | 30.19% |
| doucan and bathing cap | 27.36% | 3.77% | 4.72% | 2.83% | 61.32% |
| zucchini and vulture | 85.85% | 0.94% | 1.89% | 1.89% | 9.43% |
| jackfruit and thresher | 79.25% | 0.94% | 3.77% | 1.89% | 14.15% |
| CD plaer and beer glass | 53.77% | 10.38% | 10.38% | 1.89% | 23.58% |

presented two prompts alongside their corresponding images. The study was aptly titled, "*Given the textual concepts of two different objects, please select which of the following options creatively combines the two objects in terms of novelty, surprise, and artistic value*". We gathered responses from 106 participants who diligently completed our user study, and their decisions for each subject are presented in Table 5.

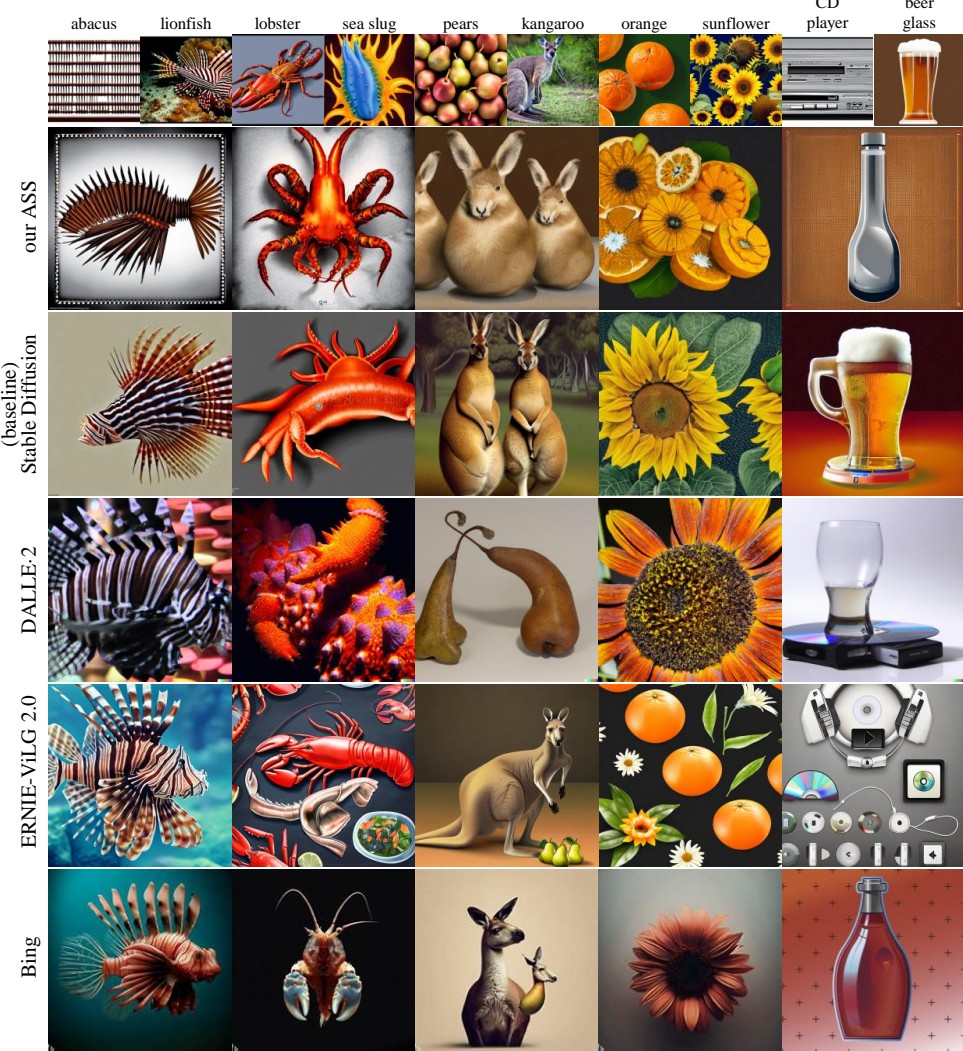

Figure 11: More visualization results of User Study.

# D PARAMETER ANALYSIS

Here, we showcase sampled images generated with varying values of $\theta$, $\overline{\alpha}$, and $\overline{\beta}$, as illustrated in Figures 12, 13, and 14.

In terms of the parameter $\theta$'s setting, our evaluation method is adept at selecting the most creative images from diverse distributions. This is made possible by dynamically determining $\overline{\alpha}$ and $\overline{\beta}$ based on these different distributions. However, this dynamic determination process is time-consuming. To streamline and expedite this process, we introduce a strict threshold for $\theta$ before proceeding to the overall ranking. This threshold effectively screens out images that require no further creative evaluation, as they exhibit evident biases. These biases, at this initial stage, contribute to the absence of creativity in the images, rendering them akin to straightforward outputs stemming directly from the prompt they are biased towards. We provide some illustrative results in Figure 12.

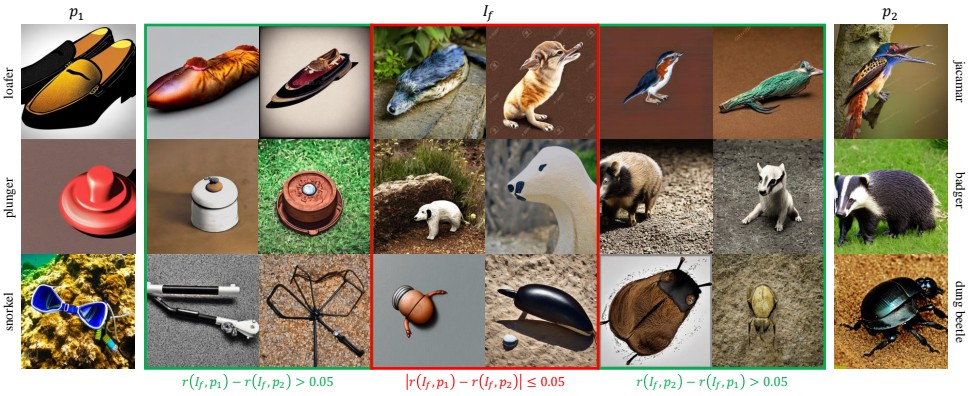

Figure 12: Visualizations with different $\theta$.

When $\overline{\beta}$ is set at 0.1, an increase in $\overline{\alpha}$ beyond our predefined value results in a biased image favoring one of the prompt pairs. Conversely, if $\overline{\alpha}$ falls below the predetermined threshold, the sampled image may appear unconventional due to the limited sampling space, as depicted in the shallow blue zone in Fig. 4. This is because the optimal sample tends to be biased for acceptability. Furthermore, when $\overline{\alpha}$ is held constant at 0.4, varying $\overline{\beta}$ can help validate the assertion that lower values of $\overline{\beta}$ result in increased urgency, whereas higher values lead to heightened confusion.

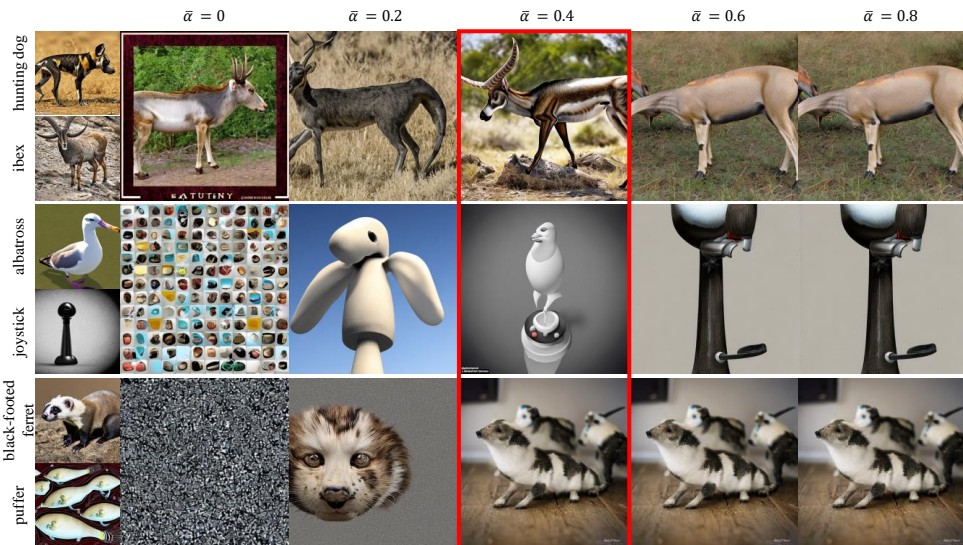

Figure 13: Visualizations with different $\overline{\alpha}$ when $\overline{\beta} = 0.1$.

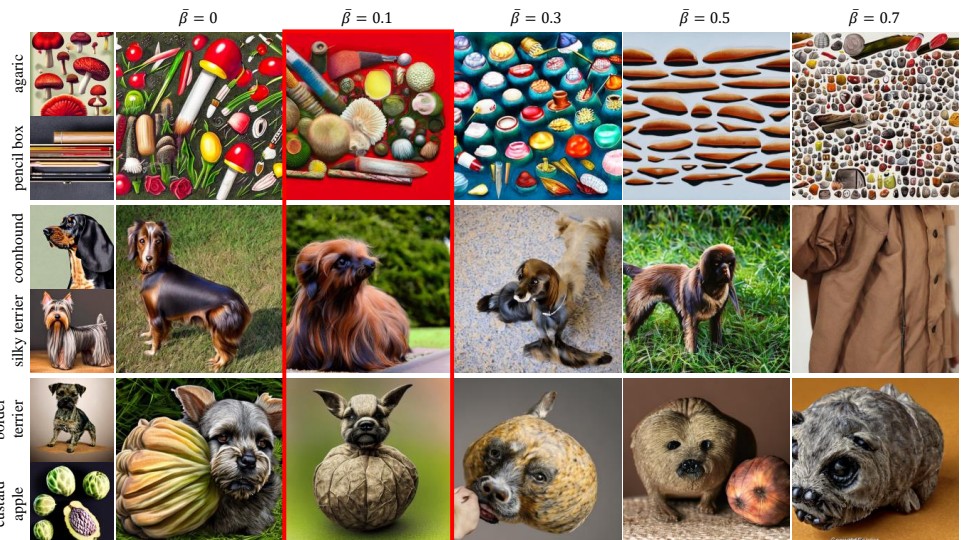

Figure 14: Visualizations with different $\overline{\beta}$ when $\overline{\alpha} = 0.4$.

# E   FAILURE EXAMPLES

Here, we illustrate instances of unsuccessful model outputs. Some prompt-pairs, as depicted in Figure 15b, prove challenging to generate novel and imaginative content. Even PickScore and HPS-v2 fail to produce satisfactory samples in such cases. While increasing the sample size may potentially address this issue, the likelihood of encountering this situation is exceptionally low, estimated at approximately 5%. Given our overall computing resources, we will refrain from excessive processing of these samples.

During the sampling stage, as observed in Figures 15a, the primary reason for subpar samples is the inadequate configuration of hyperparameters $\alpha$ and $\beta$. Despite these values being established through consensus in our experiments, they tend to align with the distribution of majority classes, neglecting the minority classes. In our upcoming research, our focus will shift towards tailoring the distribution to the most creative samples across all categories and enhancing our model to reduce the occurrence of failed samples.

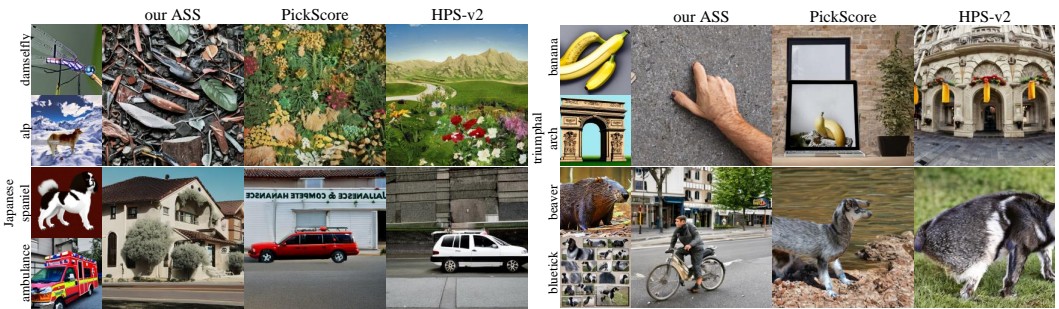

(a) Failure samples in sampling stage.    (b) Failure samples in evaluation stage.

Figure 15: Failure results

# F   MORE VISUAL RESULTS

In the following section, we delve into supplementary findings. Figure 16 illustrates a comprehensive comparison of our results with those of other T2I models. Additionally, Figure 17 showcases a collection of visually captivating and groundbreaking results.

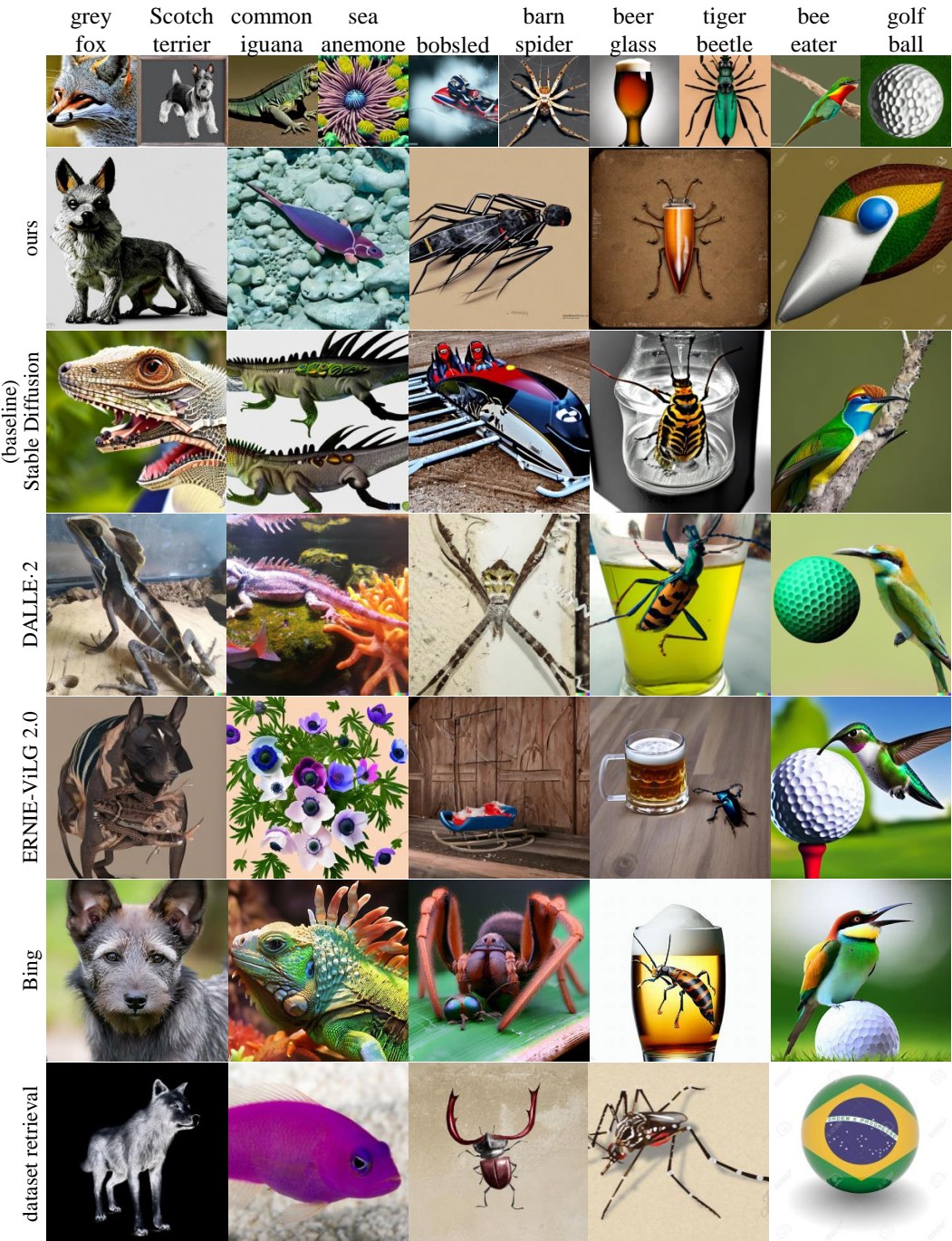

Figure 16: More visual results and comparison to other T2I models.

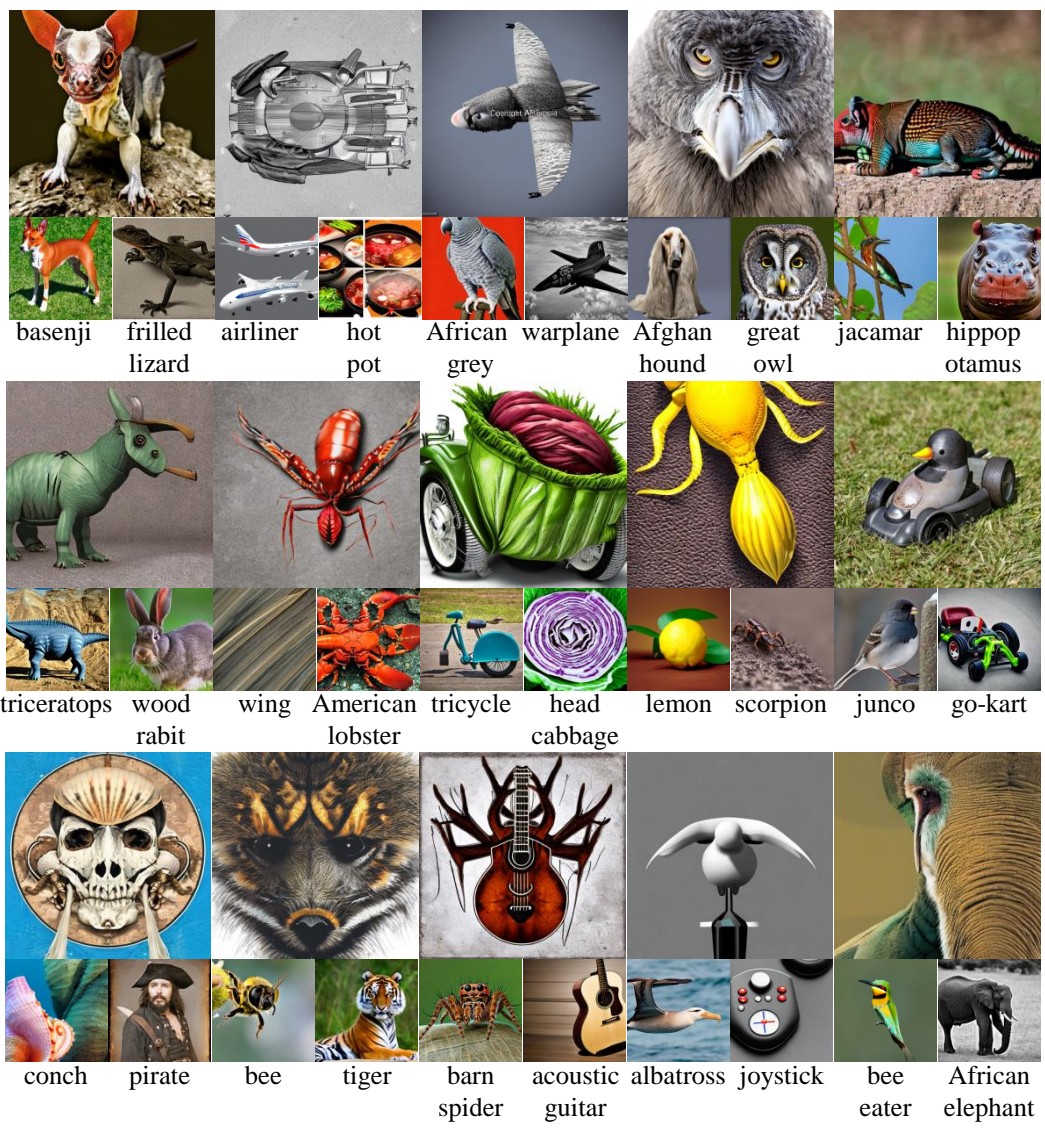

Figure 17: More visual results.

