

Figure 8: Generalizations using four different styles including *aquarelle*, *line-art*, *cartoon*, and *ink painting*. It can observe that our results are still novel and surprising.

## A  LEARNING A NEURAL SWAPPING NETWORK

In this subsection, we learn a neural swapping network to generate the meaningful combinatorial object images with different styles. The swapping part in the subsection 2.1 is rewritten as follows:

**Swapping** their column vectors by learning a neural swapping vector,

$$E_f = E_1 \text{diag}(f) + E_2 \text{diag}(1 - f) \quad \text{and} \quad f = S(\text{cat}(E_1, E_2); \psi) \in \{0, 1\}^{w \times 1}, \tag{9}$$

where $f = S(\text{cat}(E_1, E_2); \psi) \in \{0, 1\}^{w \times 1}$ is a neural swapping network from the concatenated embedding, $\text{cat}(E_1, E_2)$, to a binary output that consists of three convolutional layers and two fully connected layers with the parameter $\psi$. The architecture of the swapping network consists of three convolutional layers with a $3 \times 3$ kernel size, followed by two fully connected layers.

**Training Loss.** Using our ASS method in **Algorithm 1**, we can get the optimal combinatorial image $I_{\text{opt}}^{(p_1, p_2)}$, and then find its swapping vector $f_{\text{opt}}^{(p_1, p_2)}$. Based on the neural swapping network $f = S(\text{cat}(E_1, E_2); \psi)$ in Eq. 9 and the optimal swapping vector $f_{\text{opt}}^{(p_1, p_2)}$, the training loss is defined as:

$$L = \frac{1}{|\mathcal{P}|} \sum_{(p_1, p_2) \in \mathcal{P}} \|f_{\text{opt}}^{(p_1, p_2)} - S(\text{cat}(E_1, E_2); \psi)\|_2, \tag{10}$$

where $\mathcal{P}$ is a set of the prompt pairs $(p_1, p_2)$ corresponding to the text pairs $(t_1, t_2)$, and $|\mathcal{P}|$ represents the cardinality of the set $\mathcal{P}$. To ensure training stability, we employed the RMSprop (24) optimizer.

**Generalizations using different styles.** To evaluate the generalizations of our model, we expand its capabilities to four additional styles: *aquarelle*, *line-art*, *cartoon*, and *ink painting* for text-to-image generation. We achieve this without the need for excessive sampling and training. The results in Fig. 8, demonstrate that our model maintains its impressive creativity by generating novel species across these different styles, such as *(orangutan, king crab)*.

## B  PARAMETER ANALYSIS

In this subsection, we will demonstrate the sampled images using different values of $\theta$, $\overline{\alpha}$, and $\overline{\beta}$, as shown in Figs. 9, 10 and 11. Regarding the setting of $\theta$, our evaluation method can select the most creative images from different distributions because our $\overline{\alpha}$ and $\overline{\beta}$ are dynamically determined based on different distributions, which is a time-consuming process. Therefore, to reduce the time

required for this process, we set a hard threshold $\theta$ before entering the overall ranking. This value can preliminarily filter out images that do not need to undergo creative evaluation because these images have obvious biases, and the biases at this stage directly lead to the lack of creativity in the images, making them appear as direct outputs of the prompt they are biased towards. We demonstrate some results if Fig. 9.

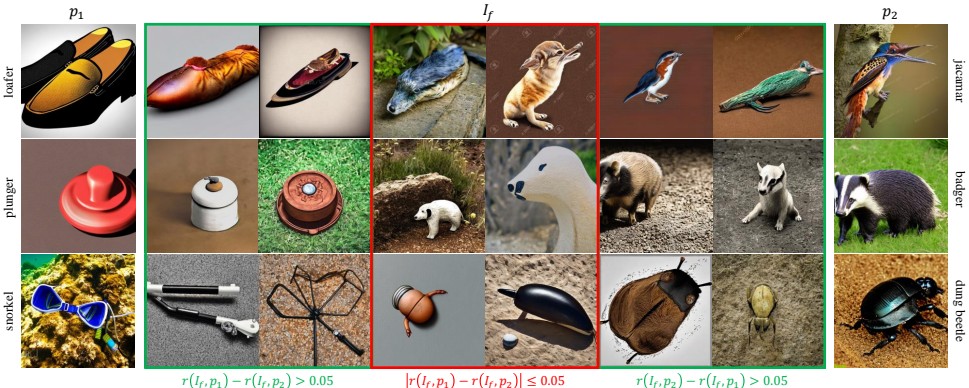

Figure 9: Visualizations with different $\theta$.

It can be observed that when $\overline{\beta}$ is fixed at 0.1, if increasing $\overline{\alpha}$ beyond our set value results, the sampled image will be a biased image towards one of the prompt pairs. Conversely, when $\overline{\alpha}$ is lower than the set value, the sampled image may appear unusual due to the narrow sampled space (as seen in the shallow blue zone in Fig. 4), that is because the best sample will be sifted for its acceptable bias. Furthermore, with $\overline{\alpha}$ fixed at 0.4, varying $\overline{\beta}$ can verify the claim that lower values of $\overline{\beta}$ lead to more emergency, while higher values lead to more confusion.

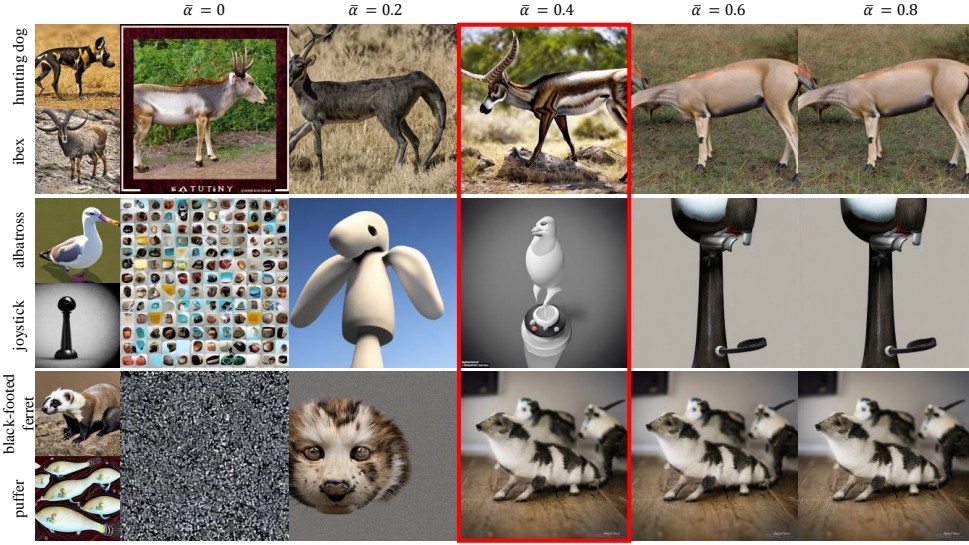

Figure 10: Visualizations with different $\overline{\alpha}$ when $\overline{\beta} = 0.1$.

## C  FAILURE EXAMPLES

In this subsection, we will present examples of failure outputs. During the sampling stage, some prompt-pairs, as shown in and 12b, are hard to compose a new and creative thing, and even PickScore and HPS-v2 also can't get satisfactory sample, and if raising the amounts of samples may fix this problem, However, the probability of this situation occurring is extremely small, accounting for only about 5%. Considering the overall computing resources, we will not overly process these samples. During the sampling stage, as shown in Figures 12a, the main reason for failure samples is the

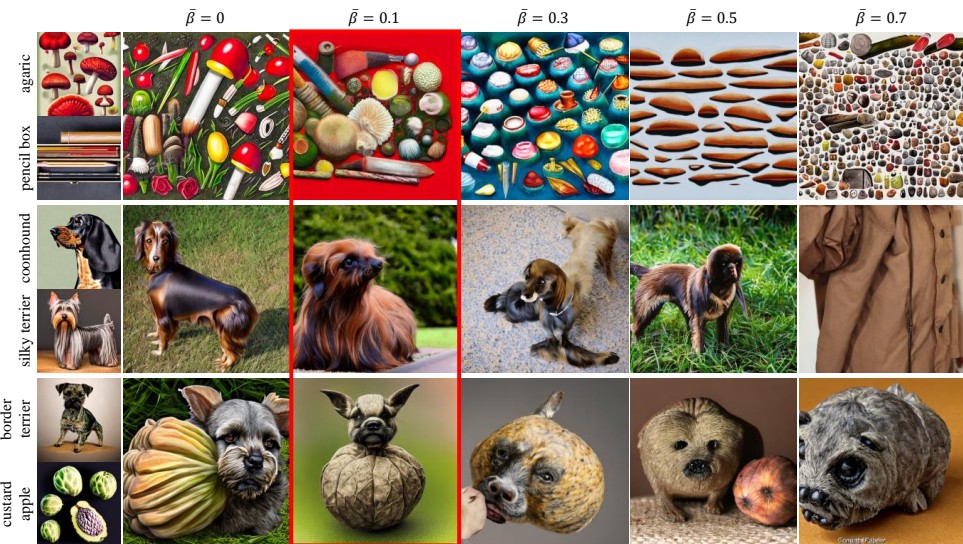

Figure 11: Visualizations with different $\overline{\beta}$ when $\overline{\alpha} = 0.4$.

improper setting of the hyperparameters $\alpha$ and $\beta$. Although these two values were determined by consensus in our experiments, they tend to match the distribution of majority classes, while ignoring minority classes. In our future work, we will focus on fitting the distribution of the most creative samples for all categories and improving our model to minimize the number of failed samples.

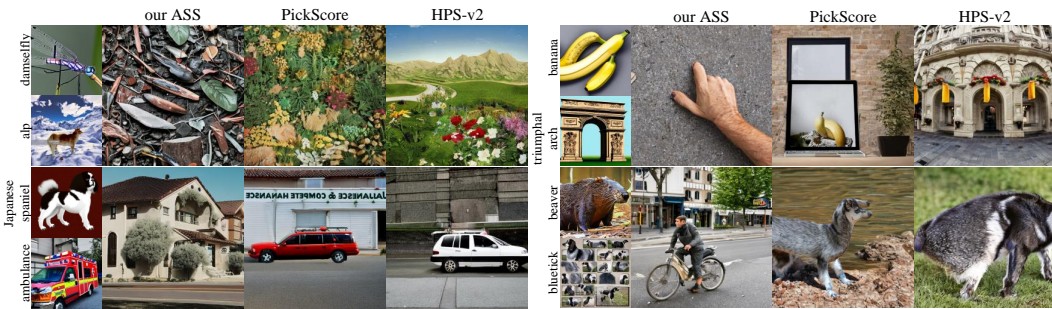

(a) Failure samples in sampling stage.   (b) Failure samples in evaluation stage.

Figure 12: Failure results

# D   MORE VISUAL RESULTS

We demonstrate more results in Fig. 13.

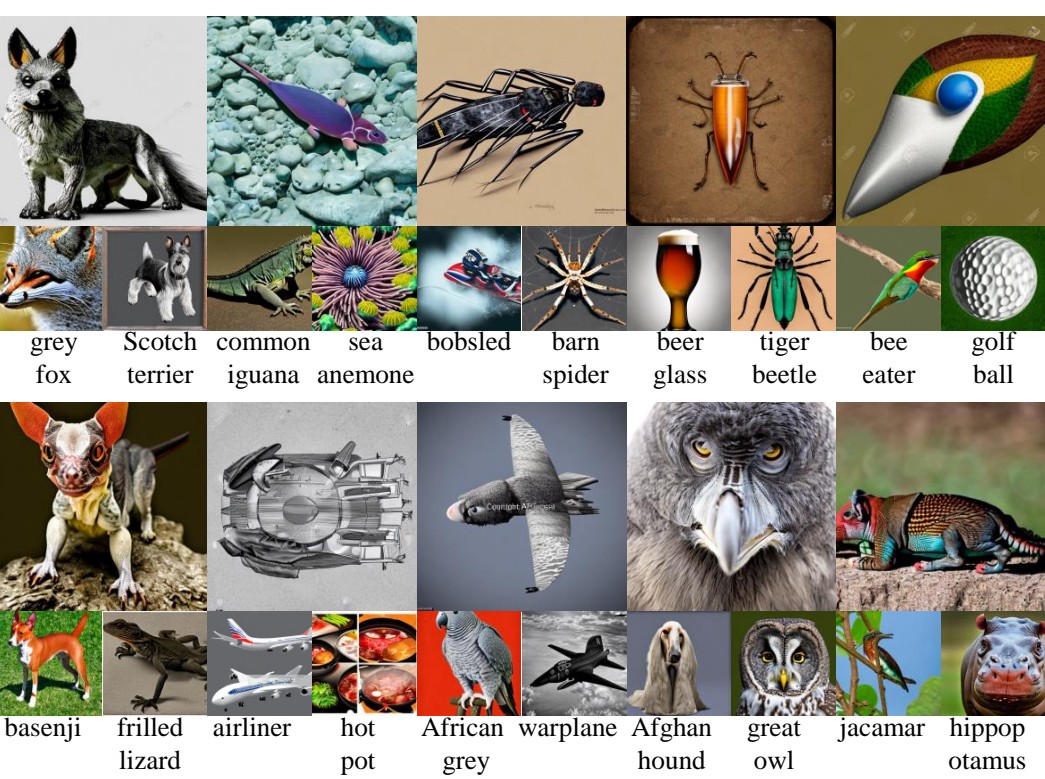

Figure 13: More visual results.