# OpenReview forum: "Amazing Combinatorial Creation: Acceptable Swap-Sampling for Combinatorial Text-to-Image Generation"
_ICLR.cc/2024/Conference — ICLR 2024 Conference Withdrawn Submission_

### Official Review · Reviewer_EqW9 · 2023-10-30

**Soundness:** 3 good
**Presentation:** 3 good
**Contribution:** 3 good
**Rating:** 6
**Confidence:** 4

**Summary:**

This paper presents a new method, named acceptable swap-sampling, to generate a combinatorial object image. Different from previous works, the proposed method uses a swapping mechanism to exchange the column vectors of two text embeddings to build a new one.  Experiments show that the proposed method generates images with better quality compared to previous works. Some numerical results and user study further support this.

**Strengths:**

- The idea of this paper is interesting. Mixing the two text embeddings to generate a new one is a novel method as far as I know.

- The presentation of this paper is good. It is easy to understand the proposed method.

- Experimental results are good. Compared to previous works, this paper receives better image generation results.

**Weaknesses:**

- It seems that the abstract is too long. The authors clearly describe the proposed method. I think there is no need to introduce the method so concretely. It would be better if it is trimmed a little bit. As such, more space can be used to add more experimental results, for instance, failure case analysis.

- Have the authors analyzed in what cases the users prefer to select other methods? It seems that the authors just simply report the numbers but further analysis is not provided.

- The authors claim that the Segment Anything Model is used in the proposed method.
  - Does this model help better generate high quality images?
  - If so, how does this model help? I cannot find relevant content to describe this.
  - In addition, does this model influence the inference speed?

- I also think more results without swapping should be added for visualization. I would like to see how the swaping mechanism works but this is done well at present.

**Questions:**

Have the authors attempted to train the models on large-scale dataset?

---

### Official Review · Reviewer_2f2Z · 2023-10-31

**Soundness:** 3 good
**Presentation:** 3 good
**Contribution:** 1 poor
**Rating:** 5
**Confidence:** 4

**Summary:**

This work tackles the creative combination of two concepts for text2image diffusion models. It proposes a latent swapping method and sampling technique to select an image from a set of initially generated images using a hybrid of two concepts.

**Strengths:**

- The paper is well-structured, and ideas are clearly explained with nice visualizations.
- The user study suggests that the presented method generates images that are more favoured by humans in terms of creativity and surprise effect.

**Weaknesses:**

- The hyperparameters are tuned using HPSv2. Doesn't this imply that the metrics used for comparison in Table 1 are directly optimized?
- How many images were generated using the baselines during evaluation? Given that the proposed method initially samples many images, the baselines should similarly be evaluated by first generating a candidate set and then combining it with PickScore or HPS-v2 to select the optimal image.
- The combination of two concepts into a surprising new image is an artistic endeavour and very subjective. While the user study indicates that the presented method is favoured more often, the baselines did not have the same chance (see point above).
- Furthermore, the technical contribution of the presented approach is limited. The equations to select the acceptable sampling region based on CLIP distance are tuned using existing method HPSv2.
- Small note: the paper states that 5k prompts were randomly selected from the combination of two random concepts (out of 1k imagenet classes), which is pretty small. How does this random selection influence the results and hyperparameters? An analysis on either a much higher prompt pool or several cross-validation sets is necessary for better judgment.

**Questions:**

- Can the authors explain why MagicMix results are so different between Fig 9 and 10? Fig. 10 results are very appealing, while Fig. 9 shows barely any difference.

---

### Official Review · Reviewer_nzuv · 2023-11-06

**Soundness:** 3 good
**Presentation:** 3 good
**Contribution:** 2 fair
**Rating:** 3
**Confidence:** 4

**Summary:**

This paper presents a novel method called acceptable swap-sampling for generating creative combinatorial object images from multiple textual descriptions. The proposed swapping mechanism constructs a new embedding by exchanging column vectors of two text embeddings, which is then used to generate a combinatorial image using Stable Diffusion. An acceptable region is designed by managing suitable CLIP distances between the new image and the original concept generations, making it more likely to accept high-quality combinations. The authors also employ a segmentation method to compare CLIP distances among the segmented components and select the most promising image from a sampled subset. Experiments are conducted on text pairs of objects from ImageNet, and the results show that the proposed method outperforms recent techniques.

**Strengths:**

1. The authors propose an interesting approach to stimulate the creativity of generative models for generating combinational images.
2. The writing is clear and easy to follow.
3. Extensive experiments are conducted and many examples are shown to validate the effectiveness of the proposed approach.

**Weaknesses:**

1. The controllability of the proposed approach is limited. It can only combine the concepts of two words but cannot control which information to use from the first image and which information to use from the second image. The model is generating some random variations without controllability. So the method can only be used to proof the creativity and imagination of generative models, but cannot be put into real usage scenarios where users always want to have detailed control of the generated images.
2. The proposed approach can only generate combinational images from two text prompts, but cannot generate combinational images from two input images.
3. There's no clear explanation of why this method works and what really happens when we swap the columns of features.

**Questions:**

Please refer to the weakness part.

---

### Official Review · Reviewer_Q86Q · 2023-11-07

**Soundness:** 3 good
**Presentation:** 2 fair
**Contribution:** 2 fair
**Rating:** 6
**Confidence:** 4

**Summary:**

This paper proposes a simple-yet-effective method, which can generate novel image based on objects contained in provided images.

**Strengths:**

1. The proposed method is simple and straight-forward. No training is needed, which means the proposed method can be applied to arbitrary pre-trained text-to-image diffusion model;

2. Good experiment results are obtained according to the paper;

**Weaknesses:**

1. The major concern is about the experiment. Due to the fact that the proposed method actually filters the generated images, it is unfair to directly compare with generated results from pre-trained models. Can the author provide the results of pre-trained models under the same setting? In other words, generate the same number of images with pre-trained models and use the same strategy to select the best generation. Results like this will show the effectiveness of both swapping mechanism and filtering criteria;


2. Because the results are generated by swapping CLIP features and filtered based on CLIP similarity, it is not surprising that the proposed method leads to better quantitative results in terms of CLIP similarity. Can the author provide results of similarity evaluated by DINO? Which can better illustrate similarity between the fine-grained details.


3. No results on images which contain multiple objects are shown.  For example, what will a combination of an image of a person riding a bike and an image of a monkey riding a horse lead to?

**Questions:**

1. Are the hyper-parameters sensitive to different object classes? I.e. Do the optimal hyper-parameters have similar values when one wants to generate combinational images for pairs of animal-animal, plant-plant, animal-plant, etc?